# Smart Spread Spectrum Modulated Tags for Detection of Vulnerable Road Users with Automotive Radar

**DOI:** 10.3390/s23052730

**Published:** 2023-03-02

**Authors:** Antonio Lazaro, Marc Lazaro, Ramon Villarino, David Girbau

**Affiliations:** Department of Electronics, Electrics and Automatic Control Engineering, Rovira i Virgili University, 43007 Tarragona, Spain

**Keywords:** Doppler radar, modulated backscatterer, automotive radar, V2X, V2I, RFID, pedestrian detection, millimeter band

## Abstract

In recent years, there has been a significant increase in the number of collisions between vehicles and vulnerable road users such as pedestrians, cyclists, road workers and more recently scooter riders, especially in urban streets. This work studies the feasibility of enhancing the detection of these users by means of CW radars because they have a low radar cross section. Since the speed of these users is usually low, they can be confused with clutter due to the presence of large objects. To this end, this paper proposes, for the first time, a method based on a spread spectrum radio communication between vulnerable road users and the automotive radar consisting of modulating a backscatter tag, placed on the user. In addition, it is compatible with low-cost radars that use different waveforms such as CW, FSK or FMCW, and hardware modifications are not required. The prototype that has been developed is based on a commercial monolithic microwave integrated circuit (MMIC) amplifier connected between two antennas, which is modulated by switching its bias. Experimental results with a scooter, under static and moving conditions, using a low-power Doppler radar at a 24 GHz band compatible with blind spot radars, are provided.

## 1. Introduction

As vehicles become increasingly smarter, vehicle networking and communications research is gaining popularity around the world [1,2,3]. The main aim of vehicular networks is to improve road safety. In addition, the provision of effective traffic management and infotainment applications is useful to avoid traffic congestion, which is another key area of concern in vehicular networks [4]. In recent years, automotive radar sensors have begun to be incorporated in a standardized way, even in mid-range cars. A number of cars already incorporate automatic braking and pedestrian safety functions [5]. Radar is the most suitable technology for this purpose compared to other technologies such as Light Detection and Ranging (LIDAR) [6] or camera-based sensors [7] because it is not affected by weather or light conditions [8] and is the best solution for measuring relative velocity. Therefore, automotive radar is expected to continue being a fast-growing application that can improve driving safety. This type of radar plays a crucial role in helping drivers to maintain safe distances for avoiding collisions [9]. Indeed, radar is becoming a key sensor for advanced driver assistance systems (ADAS) for future generation of autonomous vehicles. These vehicles will need sensors that sense the environment in real time. To this end, radar is a complementary technology to others such as those based on computer vision or LIDAR.

Automotive radars can be classified as a function of the type of waveform they use [10]. Simple Continuous Wave (CW) radar allows the determination of the velocity of the target based on the Doppler frequency shift. The resolution of the velocity measurement is inversely proportional to the time during which the radar scans the target. Although this radar is useful for movement detection applications, CW radars cannot provide target range information. Frequency-modulated continuous-wave radars (FMCW), also known as Linear Frequency Modulation (LFM) radars, can simultaneously measure both the target range and its relative velocity. FMCW radars are widely used in automotive applications. The range resolution is inversely proportional to the sweep bandwidth. Therefore, the greater bandwidth available in the 77 GHz band compared to that available in the 24 GHz Industrial, Scientific and Medical (ISM) band allows for improving range resolution. In Frequency Shift Keying (FSK) and Stepped Frequency Continuous Wave (SFCW) radars, the transmitted frequency varies in a discrete manner instead of varying continuously as in the FMCW radar [11]. The FSK radar interrogates using two frequencies alternately. It can determine velocity as in CW radars but also range. The range resolution depends on the frequency step and requires simpler hardware structure and signal processing techniques than FMCW radars. Combinations of the different waveforms are also proposed, such as multislope FMCW waveforms to avoid ghost targets [11] and interferences between multiple radars [12] or alternate pulses of CW and FMCW chirps to improve velocity and range measurement accuracy [13]. Multiple-input multiple-output (MIMO) radar is an advanced type of phased array radar that combines the last types of waveforms and signal processing techniques, which also allows for the estimation of the angle. Automotive radars typically use MIMO radars based on virtual arrays, which are more compact than classic MIMO [10]. Radars at millimeter-wave (mm-wave) band allow for reducing the size of the front end because they work at high frequencies. In recent years, significant progress has been made in the integration of 24 GHz radar sensors into automobiles. Radar sensors at 24 GHz band are less expensive compared to those based on the 77 GHz technology. Therefore, 24 GHz radar technology offers a cost-efficient solution for ADAS (Advanced Driver Assistance Systems) such as blind spot detection and short-range radars [14] (Figure 1). Blind spot detection is achieved using two corner radar sensors hidden in the rear bumper, one on the left and one on the right.

The concept of vulnerable road users refers to pedestrians, persons living with disabilities, the elderly, cyclists, people that work in the road and more recently scooter riders who are easily injured in a space dominated by cars, both on roads and on city streets. In addition, within the framework of the fight against climatic change and the reduction of emissions, cities are promoting the adoption of cycling infrastructures [15]. This policy change is reflected in the increase in both bicycle sharing and the number of bicycles and scooters. In most cities, some vehicle lanes have been removed to become bicycle lanes. An important number of vehicle collisions with vulnerable road users occur in places with low light, either at night or due to adverse weather conditions [9,16]. The ability to detect users by means of a vehicle-equipped radar strongly depends on the power backscattered by these users, which is also proportional to their Radar-Cross Section (RCS). The RCS is a complex magnitude that depends on the orientation, polarization, size, and materials [17]. The RCS of a pedestrian at 24 GHz is around −11 dBsm [18] and is affected by his position, clothing or other high reflective objects (jewelry, mobiles, etc). This value is significantly lower than the one of a standard car (18 dBsm) or a motorcycle (5–8 dBsm) [19]. Recent advances in mm-wave technology open the door to new radar bands at 77 GHz, which achieve better range resolution due to the larger bandwidth available with respect to radars at lower frequency bands. In addition, the RCS of elements on the road is also higher at 77 GHz than in the 24–26 GHz band (on the order of 5–7 dB [20]), but the problem of detecting objects such as pedestrians or bikes with low RCS is still unsolved. In addition, the speed of pedestrians and bikes is very small (<15 km/h for pedestrians). Therefore, these vulnerable road users can be confused with clutter due to the presence of large static objects such as the soil ground or metal parts such as traffic signs and crash barriers. This situation is common in cyclists or scooters riding on roadsides, intersections with bike lanes (Figure 2), and workers doing road maintenance or near heavy machines (Figure 3). It is in these situations, where the safety of the pedestrians, workers or cyclists would be greater if they were equipped with a modulated transponder that responded to short-range radar installed in vehicles.

### 1.1. Related Works

This paper studies the feasibility of using tags to identify and improve the detection of vulnerable road users by means of CW radars. The main idea is based on the use of tags or signals that are visible within the wavelength range of the radar, in the same way as it would be carried out in the areas of the spectrum corresponding to the visible, by using reflectors, lights, or blink lights. On the other hand, it is intended to induce a ghost Doppler shift to improve the identification considering static interferences such as clutter, especially in dangerous situations such as those mentioned above.

A tag based on a rotating corner to indicate the position of pedestrians, broken-down cars or workers on the road has been proposed by the authors in [21,22]. The corner reflector induces a strong retrodirective reflection which, together with the rotation, produces a characteristic spread signal in Doppler that can be recognized by automotive FMCW radars. However, this kind of system has some drawbacks such as the size and energy consumption due to both its mechanical nature and the need for a motor that allows the structure to rotate. Additionally, this device may be difficult to use on pedestrians or cyclists due to its size and weight. Therefore, to overcome these limitations, electronic tags are preferable.

In recent works [23,24,25], the authors have presented a system for vehicle-to-vehicle (car-to-car or car2car) communication at 24 GHz. A modulated backscatter (addressed also as tag) was attached to the rear of a vehicle to be detected by another vehicle’s frontal FMCW radar. The radar of the ADAS system works as a reader without the need to add any type of additional electronic or communication infrastructure. By changing the modulation frequency, the system can distinguish different events from the measured range-Doppler map (e.g., turn lights, stop, reverse direction). The modulation of the RCS allows the detection of the tag even in the presence of higher unmodulated backscatters such as the metallic car body itself. A review of some mm-wave semi-passive tag implementations can be found in [23].

Recently, environment backscatter communications are gaining great interest as low-power communication systems for the Internet of Things (IoT) applications, which use Radio Frequency (RF) signals from existing wireless sensor networks (e.g., TV, WiFi, Bluetooth, LoRa) [26,27,28]. Although Direct Sequence Spread Spectrum (DSSS) is widely extended in Code Division Multiple Access (CDMA) systems such as 3G mobile networks or in GPS (Global Positioning System), it is rarely used in backscattering communications [29,30,31] and RFID systems [32,33,34].

### 1.2. Contributions

The main contributions of this work can be summarized as follows:A radio communication between vulnerable road users and the automotive radar consisting of modulating a backscatter tag, which is placed on the user, by means of a spread spectrum sequence to improve its detectability, is proposed. The detection method uses the measured Doppler spectrum, hence the method is compatible with low-cost radars that use different waveforms such as CW, FSK or FMCW, without the need for hardware modifications;A modified and improved version of the tag designed in [23,25] is used to obtain the experimental results. Here, the tag includes a Monolithic Microwave Integrated Circuit (MMIC) amplifier, which allows for increasing the amplitude of the backscattered signal, making tag detection easier in environments with high clutter;It presents, for the first time, a spread spectrum mm-wave backscatter for radar applications, which is proposed to be a potential complement to other advanced wireless communication systems in Vehicular Ad hoc Networks (VANET) [35,36,37] and technologies based on computer vision.

The paper is organized as follows: an overall description of the system and the tag detection by using an automotive radar is presented in Section 2. Experimental results are provided in Section 3. Finally, conclusions are presented in Section 4.

## 2. System Overview

### 2.1. Tag Design

An improved version of the mm-wave tag designed for car-to-car communication with an FMCW radar in [23] is used here to obtain experimental results. Details of the designed tag were recently presented in [25], where the tag is employed as a spoofing device for FMCW radars. The main characteristics are summarized below. The block diagram of the modulated tag, which is used as a Radar Target Enhancer (RTE), is shown in Figure 4. It has been designed using Commercial Off-the-Shelf (COTS) components and consists of an amplifier connected between two antennas. The incoming signal from the radar is received by one of the antennas; it is amplified and retransmitted through the other one. The amplifier consists of two low-noise amplifiers (LNA) connected in cascade to increase the gain. The LNAs are two Silicon Germanium (SiGe) MMIC (Monolithic Microwave Integrated Circuit) amplifiers (model LNA_24_04 from Silicon Radar GmbH). Each amplifier achieves a typical gain of 17 dB between 24 GHz and 29 GHz and only consumes 5.6 mA at 3.3 V. The modulation of the radar cross section (RCS) uses a power-down control pin, which is connected to the modulation signal. It is used to enable/disable the amplifiers, controlling the backscattered amplitude. Thus, there is no need to include an expensive mm-wave modulator into the system since the modulation of the RCS is carried out by switching the supply voltage of the amplifier. A photograph of a prototype of the tag is shown in Figure 5. It is manufactured on Rogers 4003 substrate (dielectric constant $\epsilon_{r}=3.54$ and height *h* = 16 mil). Microstrip rectangular patch antennas have been designed with two 3D-printed dielectric lenses on top (Figure 6) to increase their gain up to 17.5 dB. The lenses’ prototypes are manufactured with polylactic acid (PLA), whose complex dielectric permittivity has been characterized at a millimeter band in [38], obtaining a dielectric constant of 2.55 and a dissipation factor of 0.02 at 24 GHz. To improve matching, a tapered access line is included. The antenna covers the 24–24.25 GHz ISM band and has been simulated using Ansys HFSS. In order to protect the electronics, the tag and both lenses are inserted into a PLA box made with a 3D printer. Details of the dimensions and measurements of the antenna can be found in [25].

The frequency-configurable Pulse Width Modulation (PWM) output of a low-cost microcontroller is used to modulate the tag signal. In the experimental setup, an Atmel ARM Cortex-M0 ATSAM21G18A-MU microcontroller (Microchip Technology Inc., Chandler, AZ, USA) operating at a clock frequency of 48 MHz, integrated into a Seeduino XIAO module, was used. However, other microcontrollers that have PWM outputs capable of generating square wave signals up to a few kHz can also be chosen for this purpose.

Two tags should be installed at least, one in the front and one in the back, in order to overcome the blocking effect caused by body attenuation at 24 GHz. These tags could be integrated into the lights and reflectors of the scooters or bicycles, or into users’ helmets or backpacks.

### 2.2. Spread Spectrum Modulation

The incoming signal from the radar $s_{in}\left(t\right)$ is modulated by a square wave generated by a low-power generator using the PWM output of the microcontroller (see Figure 7). This signal is a train of square-wave pulses of duration τ and repetition period $T_{m}$:(1)$$PWM\left(t\right)=m\left(t\right)\sum_{n=0}^{\infty }p\left(t-nT_{m}\right)$$
where p(t) is a square-wave pulse of duration τ and unit amplitude.

The backscattered signal can be written as:(2)$$s_{b}\left(t\right)=A_{0}\cdot s_{in}\left(t\right)+A_{s}\cdot s_{in}\left(t\right)\cdot m\left(t\right)\cdot \sum_{n=0}^{\infty }p\left(t-nT_{m}\right)$$
where $A_{0}$ is the amplitude of the backscattered field associated with the structural mode, and $A_{s}$ is the amplitude of the differential or antenna mode including the amplifier gain. In (Equation 2), $f_{m}=1/T_{m}$ is the modulation frequency of the square-wave pulse, and m(t) is the modulating sequence. When m(t) is in high-state, the PWM output modulates the tag signal, whereas the amplifier is off when m(t) is in low-state. As the square wave is a periodic function, it can be expanded into a Fourier series:(3)$$p\left(t\right)=a_{0}+\sum_{n=1}^{\infty }a_{n}cos\left(n2\pi t/T_{m}\right)$$
where $a_{n}$ are the Fourier coefficients:(4)$$a_{n}=\left\{\begin{array}{cc} \tau /T_{m} & ,n=0\\ \frac{2}{n\pi }sin\left(\frac{n\pi \tau }{T_{m}}\right) & ,n>0 \end{array}\right.$$

Assuming a 50% duty-cycle, only the odd harmonics are present. The term n=0 corresponds to the unmodulated term which overlaps the structural mode of the tag. A receiver tuned at the fundamental frequency can demodulate the information. The higher harmonics’ amplitudes decrease quickly with the index. The spectrum is schematically shown in Figure 8. The backscattered power at $f_{c}\pm f_{m}$ is proportional to the differential radar cross section, $RCD_{dif}$ [39]:(5)$$RCS_{dif}=\frac{\lambda^{2}}{4\pi }G_{t,in}G_{a}G_{t,out}M$$
where λ is the wavelength, $G_{t,in}$ and $G_{t,out}$ are the gains of the input and output antennas of the tag, respectively, $G_{a}$ is the amplifier gain, and *M* is the modulation factor. *M* is given by the square of the Fourier coefficient at the modulation frequency offset $f_{m}$ from the central carrier. For a 50% duty cycle square waveform, it is given by $M=1/\pi^{2}$ [40]. From (Equation 5), it can be concluded that the differential radar cross-section of an active transponder is increased by an amount equal to the amplifier gain compared to a passive backscatter. Therefore, the detection of the tag is better compared to that obtained in passive backscatter.

In addition to the reflected component coming from the tag, the radar receives contributions from different objects close to it (clutter). If the first sideband is considered (*n* = 1), since the components corresponding to the highest harmonics decrease very quickly with the index *n*, the received signal (amplitude of received field) by the radar can be expressed as:(6)$$X_{R}\left(t\right)=\frac{A_{tag}}{r_{tag}^{2}\left(t\right)}cos\left(2\pi f_{m}\left(t-\tau_{tag}\right)\right)\cdot X_{T}\left(t-\tau_{tag}\right)+\sum_{i}\frac{A_{i}}{r_{i}^{2}\left(t\right)}X_{T}\left(t-\tau_{i}\right)+n\left(t\right)$$
where $X_{T}\left(t\right)$ is the signal transmitted by the radar. In the case of a CW Doppler radar, it can be written as:(7)$$X_{T}\left(t\right)=Acos\left(2\pi f_{c}t+\phi \right)$$
where *A* is the amplitude of the transmitted signal, and ϕ is a phase that depends on the initial time.

In (Equation 6), $A_{tag}$ is an amplitude coefficient that depends on the differential RCS of the tag and the attenuation introduced by the pattern diagrams of the antennas due to the relative orientation of the tag and radar antennas. The coefficient $A_{i}$ takes into account the amplitude of the reflection signals in the object that composes the clutter. The attenuation of the signal increases as the square of the distance between radar and tag, $r_{tag}$. The distances between the clutter objects and the radar are denoted by $r_{i}$. The last term in (Equation 6) is the noise n(t). The propagation delay between the tag and the radar, and between the clutter objects and the radar, are $\tau_{tag}$ and $\tau_{i}$, respectively. These delays can be calculated from the distance to the radar:(8)$$\tau_{tag}=2r_{tag}/c$$
(9)$$\tau_{i}=2r_{i}/c$$
where *c* is the wave propagation velocity assuming that is approximately equal to the speed of the light in the vacuum.

The distances to the radar can be written as a function of the initial distances, $r_{tag_{0}}$ and $r_{i0}$, and the radial relative velocity component between the tag and radar (*v*), and the object *i* and the radar ($v_{i}$):(10)$$r_{tag}\left(t\right)=r_{tag_{0}}+v\cdot t$$
(11)$$r_{i}\left(t\right)=r_{i0}+v_{i}\cdot t$$

In contrast with stationary tags typically used in backscatter communications, in the event of the tag or car moving, the received spectrum undergoes a Doppler shift that cannot be neglected and is given by:(12)$$f_{d}=-\frac{2vf_{c}}{c}=-\frac{2v}{\lambda }$$
where λ is the wavelength at the carrier frequency of the radar.

Some of the clutter contributions may have higher amplitude than the signal coming from the tag itself (e.g., reflections on the road or on other cars, etc.). The signal coming from the tag can be separated from the clutter thanks to signal modulation. Assuming an urban environment, the maximum speed is limited to 30–50 km/h. Therefore, the allocation of the modulation frequency at a region that is expected to be free from clutter contributions can help to detect the tag (see Figure 8). In addition, the information is double-sideband modulated at $f_{c}+f_{d}\pm f_{m}$. This fact can be used to distinguish a tag from other objects that may generate components in the Doppler spectrum. Note that the different components that constitute the clutter do not experience exactly the same Doppler shift since the relative velocities may be different than the moving tag.

To improve tag detection, a spread spectrum codification, which is well known for its robustness to interference, is proposed. The direct sequence spread spectrum (DSSS) technique is widely used to allow multiple users to share simultaneously the same channel in CDMA systems such as 802.11b, GPS, CDMA2000, etc. The data stream (user data) is encoded by multiplying it by a chip sequence, resulting in a broadband signal. The bit duration is the same, but the chip duration ($T_{c}$) is much shorter than the bit time ($T_{b}$), hence resulting in a larger bandwidth. The modulation sequence m[n] is obtained by multiplying the pseudo-noise (PN) sequence c[n] and the data stream d[n]:(13)m[n]=c[n]·d[n]

Note that, in (Equation 13), the sequence multiplication operation must be replaced by the XNOR operation if the sequences are coded in the binary system (0 and 1) instead of the bipolar one (−1 and 1). If the value of the sequence m[n] is 0 (or −1 in the bipolar system), the PWM frequency is set to $f_{m0}$, whereas, if m[n]=1, the PWM frequency is set to $f_{m1}$. Chip time duration is assumed to be much longer than the PWM period. By using this modulation schema, both Amplitude-Shift Keying (ASK) modulation, if $f_{m0}=0$, or Frequency-Shift Keying (FSK) modulation, if $f_{m0}\neq f_{m1}$, can be implemented.

Since the aim of this work is to reuse the radar front-end hardware, the demodulation process must be compatible with the existing radar techniques. Figure 9 shows the simplest mm-wave radar structure with one transmission channel and one receiving channel. An FMCW radar has the same block diagram, but the carrier frequency is linearly swept within the range of frequencies corresponding to the radar bandwidth. In the receiver, the backscattered signal is amplified and down-converted with an IQ mixer. The phase and quadrature outputs of the mixers are sampled using fast analog-to-digital converters (ADC). The Fast Fourier Transform (FFT) is performed in the radar processor to obtain the target velocity. In addition, in the case of FMCW, the range-Doppler map can be also obtained from a 2D FFT. In this work, the simplest case where the carrier frequency is constant is considered. The detection of a modulated backscatter transponder using an FMCW radar has been described in other work [23].

### 2.3. Tag Detection

The detection of the tag is schematically shown in Figure 10. FSK or ASK demodulation is performed from the spectrum estimation using the standard processing techniques applied in radars. The spectrum is obtained from the FFT of the IQ samples at the IF output of the radar mixer (I[n] and Q[n]). A Hamming window is used to reduce the sidelobe interference and zero padding is applied to improve the FFT frequency resolution. When the radar receives a 0 or 1 bit, it should be detected a peak at $\pm f_{m0}+f_{d}$ or $\pm f_{m1}+f_{d}$ in the spectrum, respectively. Figure 11 shows a schema of the received spectrogram (spectrum as a function of time) whether it is modulated in FSK or ASK.

A crucial point for the correct processing of the signal is determined by the separation of the reflections from the target in the Doppler spectrum. As long as the received signal that is composed of the target reflections, noise and clutter reflections is greater than a threshold, detection for the specific frequency cell will be considered. In order to determine this threshold, a one-dimensional Constant False Alarm Rate (CFAR) has been used. The detection is based on the estimation of a threshold from a set of training cells. Different CFAR processors have been proposed in the literature to perform this estimation. The well-known Cell-Averaging Constant False Alarm Rate (CA-CFAR) noise estimation is obtained by averaging the values of the cells in the training area. In the CA-CFAR, the threshold is computed by taking the average amplitude of the cell around the Cell Under Test (CUT). In this averaging process, some cells (guard cells) around the CUT are excluded to avoid the influence of the CUT level in the calculation (see Figure 12). The threshold can be implemented using a digital filter:(14)$$Z\left[k\right]={\left|X[k]\right|}^{2}⨂H\left[k\right]$$
where ⨂ denotes the convolution operator and H[k] are the filter coefficients (H[k]=1 for the training cells and zero for the others). The threshold is estimated as α times the value of the average amplitude Z[k]. A typical value for the α is 12 dB.

A one-dimensional peak search routine is used for peak detection. It should be noted that the difference between the maximum frequencies of the positive and negative sidelobes does not depend on the Doppler shift $f_{d}$ (see Figure 8). Therefore, in the case of FSK modulation, if this difference is close to $2f_{m0}$, it is decided that a 0 bit is sent. However, if it is close to $2f_{m1}$, the bit sent is a 1. In the case of ASK modulation, the procedure is easier than in FSK. A bit 1 or 0 is sent if the spectrum is respectively above or below a specific threshold.

The extracting data operation is performed by multiplying the received signal by the pseudo-noise sequence and integrating the result. This operation can be easily performed by correlating or convolving the received signal with a replica of the spread code. In low-cost radars, low-power microcontrollers with relatively low clock frequencies are used. Therefore, the number of bits per chip is limited by the time required to perform the FFT and computer operations. The immediate consequence is that there is a limitation in the transmission velocity. Another limiting factor, especially where the relative velocity is high, is the time during which the radar has the tag within its field of view due to the short range of the system. If the scanning time (the time the radar illuminates the tag) is too short, the received sequence may be cut off and not fully received.

### 2.4. Spreading Sequence

The ability of DSS signals to decode data from different users if the spread sequences (c[n]) of each of them are uncorrelated with each other is well known. This situation can occur if two tags are within the radar coverage range. The output of the demodulator can be expressed as:(15)$$s\left[n\right]=\alpha_{1}d_{1}\left[n\right]\cdot c_{1}\left[n\right]+\alpha_{2}d_{2}\left[n\right]\cdot c_{2}\left[n\right]+n\left[n\right]$$
where $\alpha_{1}$ and $\alpha_{2}$ are the amplitude for each user. It should be noted that the sequences received for each tag are not synchronized. From the radar equation, these voltage amplitudes are inversely proportional to $r_{i}^{2}$ ($r_{i}$ is the distance between each backscatter and the radar). Therefore, the nearest backscatter masks the possibility of decoding the far backscatter. This is a well-known problem in CDMA communication systems, which is solved by performing a power control to try to guarantee that the signals from different users are received at similar levels. This power control cannot be applied in a backscatter system if an uplink channel does not exist. Therefore, backscatter number 1 encoded with the code $c_{1}$ can be decoded if the peak of the autocorrelation with the code $c_{1}$ ($R_{c_{1}c_{1}}$) is greater than the peak of the cross-correlation with the code $c_{2}$ ($R_{c_{1}c_{2}}$), which is true if the ratio of distances is approximately equal to:(16)$$\frac{r_{2}^{2}}{r_{1}^{2}}\approx \frac{max(R_{c_{1}c_{2}})}{max(R_{c_{1}c_{1}})}$$

The signal-to-inference ratio (SIR) between two users with some received level is given by the main lobe level of the autocorrelation function with respect to the peak of the cross-correlation function. The SIR in dB is given by:(17)$$SIR\left(dB\right)=10log\left(\frac{max(R_{c_{1}c_{1}})}{max(R_{c_{1}c_{2}})}\right)$$

For example, if the system uses codes with a SIR of 10 dB, and assuming that the maximum distance at which a backscatter can be detected (read range) is about 10 m, the closest tag to ensure that multiuser decoding can be performed can be placed within 3.16 m of the radar. Therefore, in practice, only tags closest to the radar can be decoded using CDMA. From an application point of view, the detection of a user within the reading range should cause an alarm in the vehicle, regardless of whether there are one or two tags.

Orthogonal codes such as Walsh codes can be used. However, they require strict synchronization between the backscatter and the reader. This condition is difficult to satisfy without a control uplink channel. Another option is the use of pseudo-noise (PN) codes such as Barker codes [41], M-sequences, or Gold codes with good auto-correction properties [42]. The presence of PN sequence can be optimally detected using correlation. With a PN sequence, the highest correlation is obtained when the received and reference sequences are aligned and is close to zero if they are displaced relative to each other (the Barker codes’ sidelobe autocorrelation level is 1/N times less than that of the signal maximum [43]). This is an ideal characteristic for synchronization. Unfortunately, there are no known Barker codes with a length N greater than 13 (see Table 1). Maximum length sequences (generated with linear feedback shift registers) also have this property of displacement correlation and include sequences that are significantly longer than Barker codes. In the last case, the processing gain is a function of the length of the codes. Gold codes used in GPS are another example of those that present a low cross-correlation level. Another interesting code is the orthogonal Golay complementary sequences [44]. The sum of autocorrelation functions of the Golay codes that form a complementary set suppresses the sidelobes present in the individual autocorrelation functions. Therefore, their cross-correlation functions also sum to zero. Moreover, they are mutually orthogonal codes. Golay complementary pairs can be generated recursively following the procedure described in [45,46]. A method for designing long Barker-like codes is described in [47].

In the context of the proposed application, some practical simplifications can be considered. The aim is the detection of a vulnerable road user such as a pedestrian, a bicycle or a scooter instead of the transmission of long variable messages. In order to avoid collision between multiple backscatters, different modulation frequencies can be used. Therefore, a specific channel can be reserved for the identification of pedestrians, bikes or scooters and another for traffic signals. For each traffic signal, a different code can be employed. A traffic signal is generally grouped in the same location; therefore, a single backscatter can cyclically repeat the code associated with each traffic signal. Considering that the demodulation and correlation process will be performed in real-time using low-cost microcontrollers, the bit time will be relatively long. The presence of the Doppler effect associated with the movement of the vehicle and the tag makes it difficult to detect the latter. To increase the detection reliability and to avoid variations of the received frequency during the bit time due to Doppler shifts, a large frequency deviation ($f_{m1}-f_{m2}$) should be used in the case of choosing FSK modulation. Therefore, the use of ASK modulation is a simpler solution than FSK modulation since the detection of peaks in the spectrum is simplified and consequently less processing time is needed. For this reason, the case of ASK modulation is considered in this work.

## 3. Results

The experimental results have been obtained using a low-cost Doppler radar K-LD7 model from RFbeam Microwave GmbH. Figure 13 shows an image of the radar module including the receiving and transmitting antenna arrays. The radar works in the 24 GHz ISM band and transmits an EIRP of 6 dBm. The antenna gain is 8.6 dBi. The horizontal and vertical −3 dB beam widths (HPBW) are respectively 80${}^{\circ }$ and 34${}^{\circ }$. The radar has two receiving antenna arrays with a separation between them of *d* = 6.223 mm. This radar can operate in FSK mode, using two transmission frequencies. However, for the sake of simplicity in this work, we are going to focus on tag-to-radar c ommunication in which only one receiving channel (Doppler radar) and one transmission frequency are required. The FFT is performed from the ADC samples taken by the microcontroller included in the radar board. The Fourier samples were sent to the host computer via the USB port. The rest of the signal processing is carried out on the host computer. The signal processing in the host computer is implemented in Matlab and Python. Therefore, both PC and other platforms such as Raspberry Pi can be used as host computers.

The following figures show the experiments performed with the radar and the tag previously described, which is attached to a scooter. Firstly, some experiments have been performed without moving the tag to show the modulation capability and check the correct operation of the CFAR algorithm. They have been performed inside the hall of the laboratory separating the tag and radar 2 m from each other. Figure 14 shows the spectrogram (spectrum as a function of the time) when the tag is not modulated and when it is with a train of square wave pulses at different frequencies between 500 Hz and 3500 Hz in steps of 500 Hz. The modulation frequency is held constant for 5 s before changing to the next frequency. A Hamming window has been used to reduce sidelobes in the Fourier transform, and 256 points have been considered in the FFT computation. The maximum velocity is a configurable parameter that depends on the ADC sampling frequency (≈9 kHz), and it is set to 100 km/h for the following experiments. Therefore, the maximum modulation frequency is limited by the Nyquist theorem to halve the sampling frequency (4500 Hz with this configuration) to prevent aliasing. The noise level is similar for all frequencies in the spectrum except for those close to zero due to the leakage in the mixers. Therefore, modulation frequencies below 250 Hz should be avoided. Figure 15 shows the spectrogram after applying the CFAR processor with 20 reference cells and 6 guard cells at each side around the cell-under-test. The Fourier transform envelope of a square pulse is a sinc function (sin(x)/x) with a sidelobe level 13 dB below the main lobe. In this figure, the threshold chosen for the CFAR processor is 12 dB, in order to eliminate the higher harmonics at $n\cdot f_{m}$ that are 13 dB below the fundamental frequency level. This figure clearly shows the sideband associated with the modulation frequency of the tag ($f_{m}$). These figures show that the tag can be detected in the Doppler spectrum even though it was not moved without blind distance (minimum detection distance). In this case, the detection of the tag is especially important in applications like the one in Figure 3. It is also important when the velocity of the scooter is very low or when it is stopped in front of a traffic light.

The following figures summarize some outdoor experiments for the detection of a scooter that has a tag. The tag signal is ASK modulated at 2 kHz, and it sends data periodically every 2 s. The bit time used is 300 ms and is limited by the time required by the microcontroller to perform the FFT operations which is about 90 ms. Therefore, the number of samples per bit measured by the radar is approximately 3. Firstly, measurements at 8 m sending Barker’s codes of length 7 and 13 are shown in Figure 16 and Figure 17, respectively. These figures show the spectrogram of the measurements, the correlator and the presence detector responses, after comparing the signal at the output of the correlator with a threshold (70% of the expected maximum output of the correlator). As expected from Table 1, the correlator output for the 13-bit Barker code is about 5 dB higher than for the 7-bit Barker code; however, the length of the first code is higher.

In Figure 18, Figure 19, Figure 20 and Figure 21, the scooter is moving. The scooter passes in front of the radar periodically at different velocities and returns to the original position after traveling about 15 m. When the tag was within the radar reading range (limited to about 10 m due to the transmission power of the radar used in this work), the received data can be observed. A frequency shift appears due to Doppler. The trace at low frequencies observed in the spectrogram is caused by reflection from the metal parts of the scooter and the body. The Doppler frequency is variable and also depends on the orientation angle between the radar and the scooter (see Figure 19 and Figure 21). The Doppler shift is estimated from the peak of the spectrum in the frequency range between −1000 and 1000 Hz. Additionally, a smoothing filter is applied to reduce noise in the estimation:(18)$$f_{d}\left[k\right]=\alpha \cdot f_{d}\left[k-1\right]+\left(1-\alpha \right)\cdot f\left[max\left[\left|Y(f)\right|,\left|f\right|<f_{d,max}\right]\right]$$
where α=0.9 is a smoothing factor, and $f_{dmax}=$ 1000 Hz is the frequency range used to find the maximum of the spectrum of the CFAR processor output Y(f).

The Doppler shift estimated from (Equation 18) has the mission of defining the center of the frequency window used in the demodulator that allows for obtaining the peaks associated with the data. The Doppler frequency is positive when the scooter approaches the radar and negative when it moves away. The demodulated data that allow the identification of the scooter are obtained after first performing the correlation with the Barker code and then applying the threshold. It can be seen that the number of successful detections increases when the scooter is almost stopped in front of the radar (around 20 s in this measurement) and occurs when the tag is within the radar’s field of view. When the scooter returns to its starting position, the tag cannot be detected by the radar because the scooter driver blocks it. In these cases, the solution is to install a second tag on the rear of the scooter to allow detection. The output of the correlation presents a high level even if the received frame has been truncated due to the limited tag visibility.

In the following experiments, the radar is attached to the rear of a car (on the bumper) in order to emulate the typical location of radars for blind spot detection (see Figure 22a). The tag is installed on a scooter (Figure 22b). The following figures show the experimental results obtained when the car and the scooter are moving along the road in the same direction (Figure 23). The road has a bicycle lane similar to the one shown in Figure 2.

In the first experiments, Barker codes of length 7 are used with an interval of 1 s between codes. In Figure 24, the scooter is approximately less than 10 m away from the car; therefore, it is within the reading range all the time. As a result, all the frames were detected, even during accelerations or decelerations of the car (the estimated Doppler frequency for this measurement is shown in Figure 25). Another experiment is shown in Figure 26. The Doppler frequency for this case is shown in Figure 27. Initially, the tag is within the radar reading range, but when the car suddenly accelerates, for a few moments, the tag is out of the radar field of view as the distance between the car and the scooter is too great. After a while, the car brakes, and the tag is detected again.

Experiments with the tag sending Barker codes of length 13 have also been tested. The results are shown in Figure 28, and the Doppler frequency for this case is shown in Figure 29. In this experiment, the initial position of the scooter is practically outside the range of the radar, and, therefore, is not detected by it. However, when the car brakes because it is near an intersection, the scooter approaches the car and is detected.

The estimated Doppler shift corresponding to the low-frequency band of the spectrum is due to the contribution of all objects and clutter in the scene. Therefore, it changes in the presence of large reflectors such as another car (stopped or moving) that is within the angular width of the antennas. However, the value of the Doppler shift is only used to determine the position of the center of the window to be used in the spectrum peak searching algorithm to demodulate the bits. Since this has a maximum value of 1000 Hz, errors in the Doppler shift estimate caused by the scooter generally do not affect the result. The previous figures show the plot of the frequency of the peak detected when bit 1 is sent, superimposed with the plot of the estimated Doppler shift. In these figures, the peak frequency detected for bit 1 ($f_{m1}+f_{d}$) approximately follows the variations of the Doppler shift.

These experiments demonstrate the detection capabilities of modulated tags in dynamic environments such as urban ones when the vehicle is approaching intersections (Figure 23). To correctly demodulate the sequence, all its component chips must be received; otherwise, if a truncated sequence is received, the level of the autocorrelation peak decreases, making detection more difficult. Therefore, the radar must receive the signal backscattered by the tag for at least N·T seconds (where *N* the number of chips and *T* the chip time). The use of longer codes (e.g., Barker length 13) requires a longer tag visibility time and therefore higher latency because more time is needed to send the complete spread spectrum sequence. Due to the low bit rate associated with sampling constraints and the time required for Fourier transform calculations by the low-power microcontroller, the chip length *T* cannot be arbitrarily reduced, so shorter Barker codes (e.g., Barker length 7) are preferable. In the experiments, the reading range is limited by the transmitter power of the radar (6 dBm). The use of radars with a transmission power of 15 dBm allows the reading range to be increased to approximately 30 m, also increasing the visibility time.

To facilitate readability, Table 2 summarizes the different experiments (cases) and conditions that have been realized.

## 4. Conclusions

In this work, the feasibility of using modulated backscatter tags for the identification and detection of vulnerable road users using automotive radars has been studied. The objective pursued is the reduction of collisions caused by driver distractions, lack of visibility due to adverse weather conditions or absence of light or the presence of blind spots. The concept is evaluated in the 24 GHz band due to the higher availability of RF components in this band compared to higher automotive bands such as 77 GHz. To this end, a 24 GHz proof-of-concept transponder based on two low-power consumption SiGe MMIC amplifiers connected between two patch antennas is designed. The additional gain of the amplifiers allows for increasing the differential radar cross section, improving the detectability of the transponder in the presence of strong non-modulated clutter reflections. Spread spectrum encoding has been employed to improve the detection of the tag in real environments. A low-power Doppler radar operating in the ISM band works as a reader. The system does not require hardware modifications, and it is compatible with automotive blind spot radars. Although in the experimental setup the communication is limited by the time the tag is visible from the radar and by the low-power micro-controller capabilities, a successful detection has been achieved. Experimental results have been presented under static and moving conditions of a tag installed on a scooter. This tag could also be installed on other devices such as bikes or motorcycles, or carried by workers operating on the road for safety purposes.

## Figures and Tables

**Figure 1 sensors-23-02730-f001:**
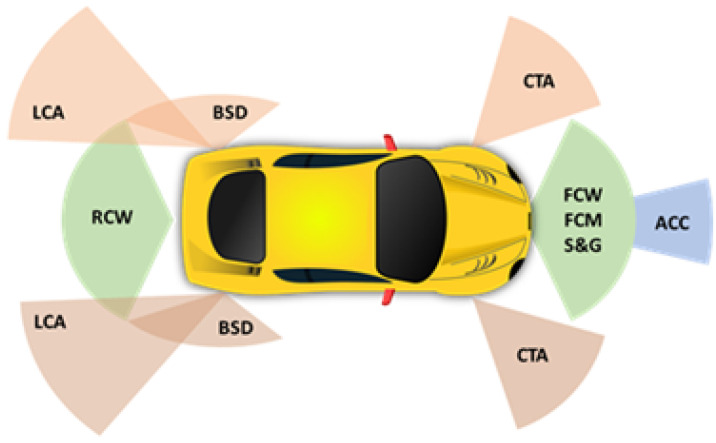
Schema of different radars installed on a vehicle and functionalities. Long range: ACC (Adaptive Cruise Control). Short/Medium range: BSD (Blind Spot Detection), LCA (Lane Change Assist), CTA (Cross Traffic Alert), FCW (Forward Collision Warning), FCM (Forward Collision Mitigation), RCW (Rear Collision Warning), S&G (Stop and Go).

**Figure 2 sensors-23-02730-f002:**
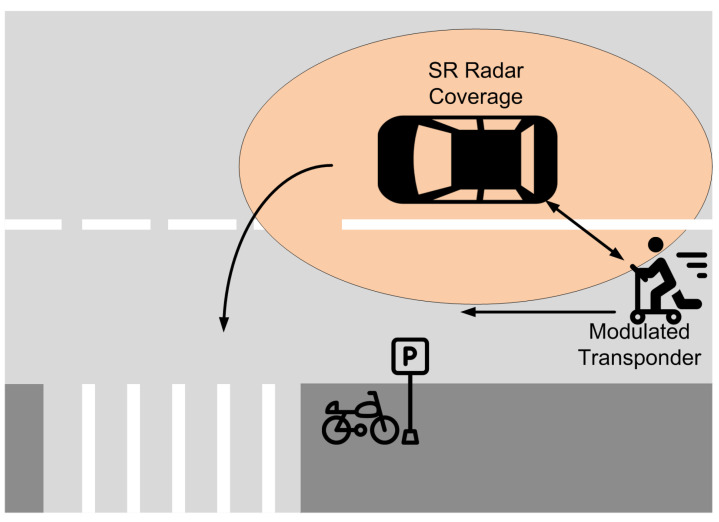
Example of use of the modulated transponder in an intersection of a vehicle with a bicycle lane.

**Figure 3 sensors-23-02730-f003:**
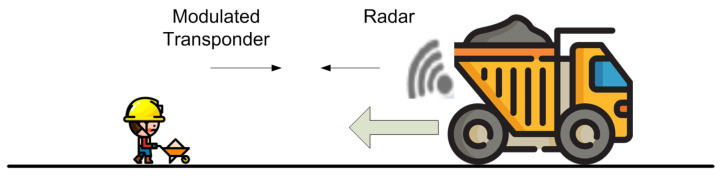
Worker carrying a transponder and a heavy vehicle moving backward equipped with a radar.

**Figure 4 sensors-23-02730-f004:**
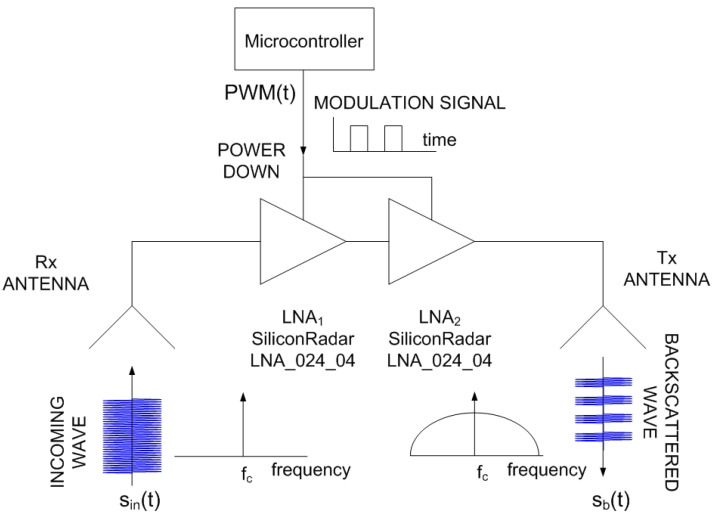
Block diagram of the tag designed as a proof of concept.

**Figure 5 sensors-23-02730-f005:**
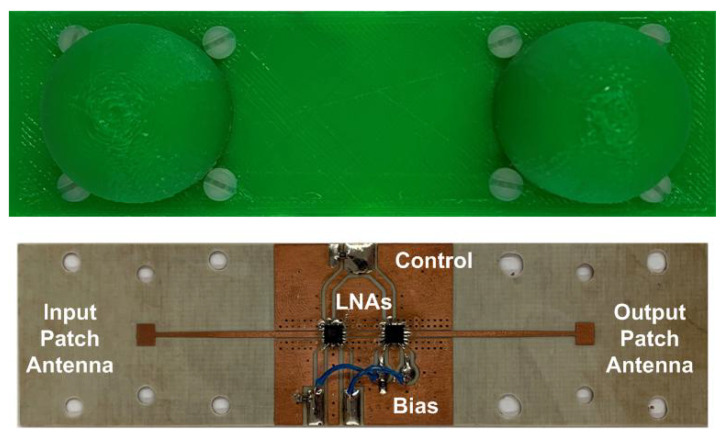
Photography of the microwave front-end of the tag prototype.

**Figure 6 sensors-23-02730-f006:**
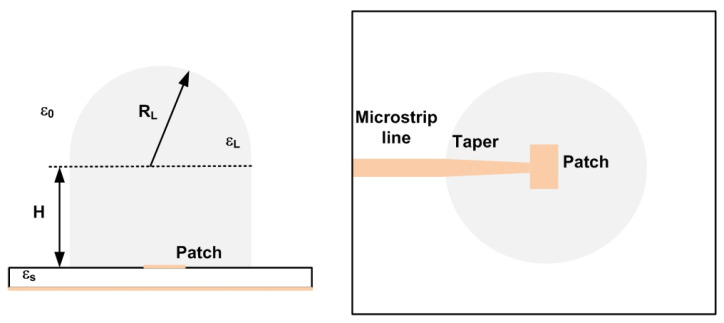
Detail of the dielectric lens antenna.

**Figure 7 sensors-23-02730-f007:**
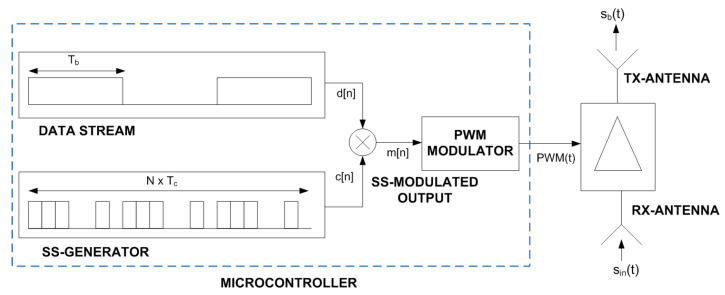
Multiplication of the data stream with a spread-spectrum sequence.

**Figure 8 sensors-23-02730-f008:**
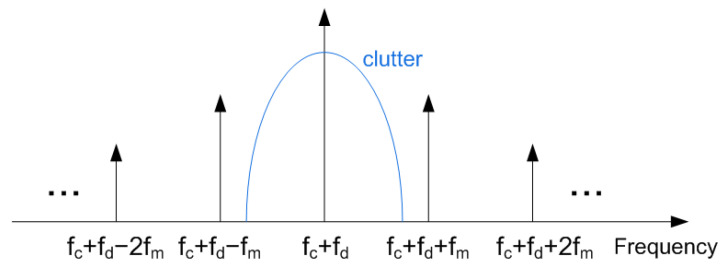
Backscattered spectrum.

**Figure 9 sensors-23-02730-f009:**
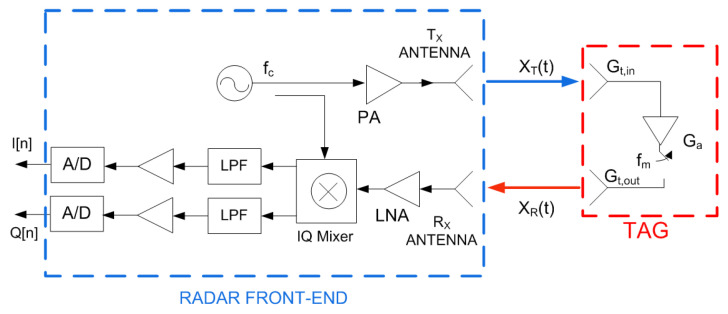
Schema of the mm-wave radar front-end.

**Figure 10 sensors-23-02730-f010:**
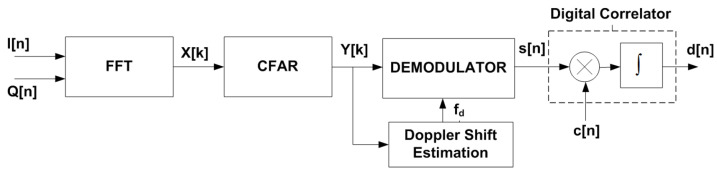
Block diagram of the despreading decoding.

**Figure 11 sensors-23-02730-f011:**
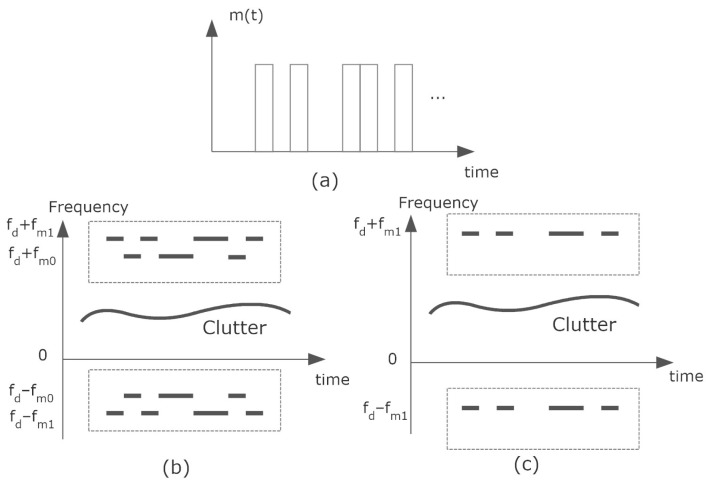
Diagram of spectrograms received (**a**) when modulating with the sequence *m*(*t*), spectrogram received if modulated in (**b**) FSK and (**c**) ASK. The dashed lines indicate the region of the window used to search for modulation peaks.

**Figure 12 sensors-23-02730-f012:**
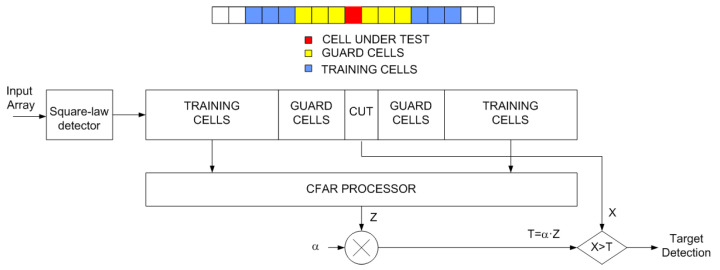
Block diagram of the CFAR processor.

**Figure 13 sensors-23-02730-f013:**
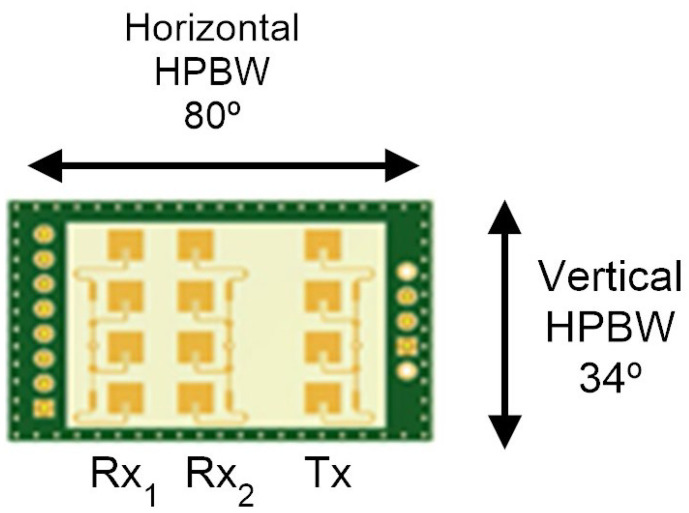
Detail of the receiving and transmitting antenna arrays and HPBW in the horizontal and vertical direction.

**Figure 14 sensors-23-02730-f014:**
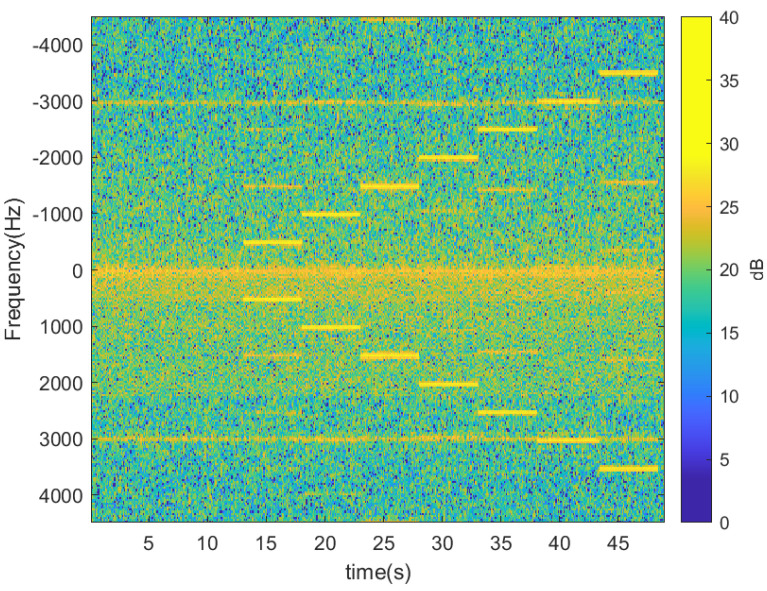
Spectrogram for different modulation frequencies with the immobilized tag located in front of the radar at 2 m.

**Figure 15 sensors-23-02730-f015:**
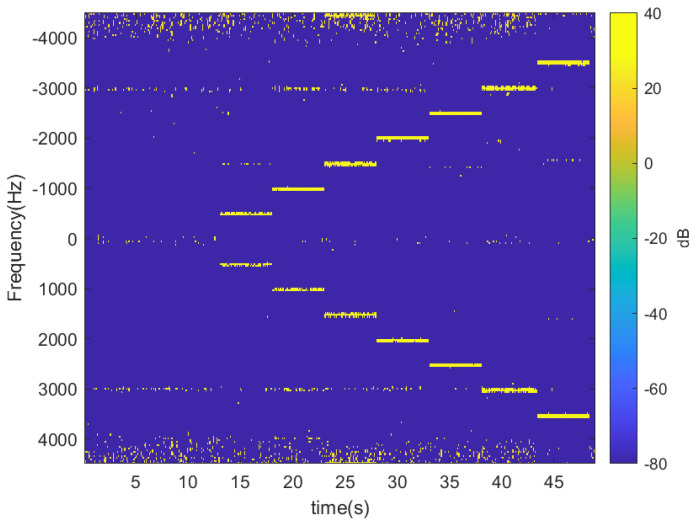
Spectrogram for different modulation frequencies with the immobilized tag located in front of the radar at 2 m after applying the CFAR algorithm.

**Figure 16 sensors-23-02730-f016:**
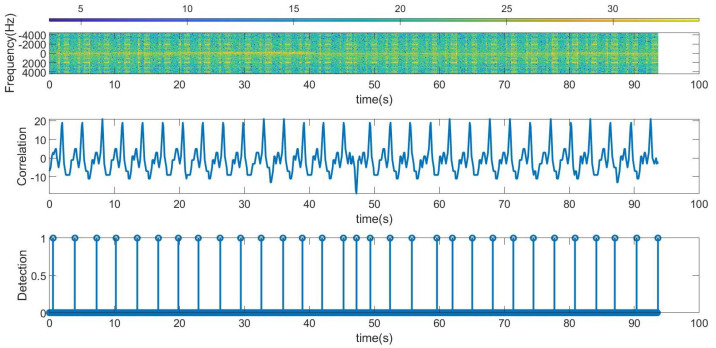
Measurement for a stationary scooter sending Barker codes of length 7 at a distance of 8 m. From top to bottom: raw spectrogram before CFAR processor, the signal at the output of the correlator and tag detection.

**Figure 17 sensors-23-02730-f017:**
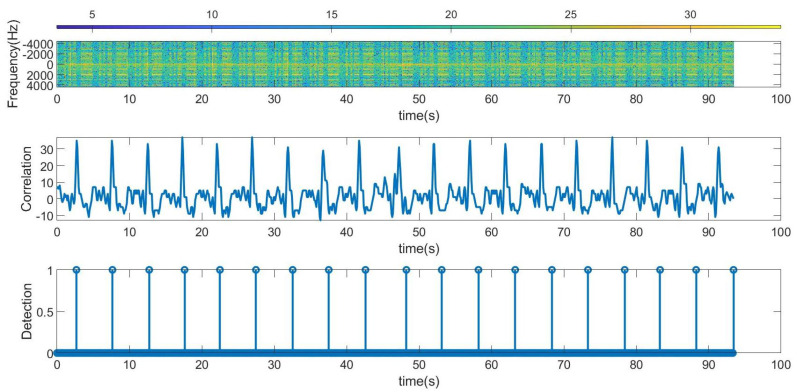
Measurement for a stationary scooter sending Barker codes of length 13 at a distance of 8 m. From top to bottom: raw spectrogram before the CFAR processor, the signal at the output of the correlator and tag detection.

**Figure 18 sensors-23-02730-f018:**
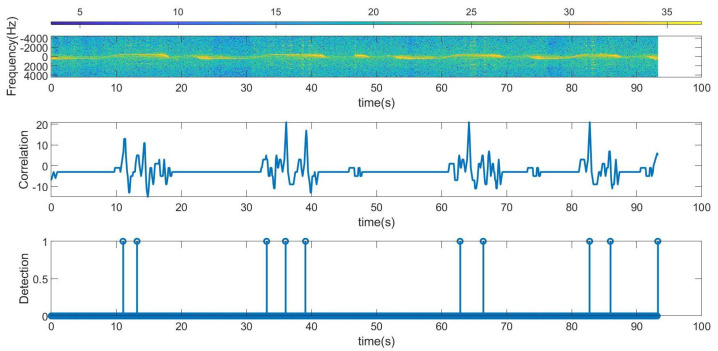
Measurement for a moving scooter sending Barker codes with length 7: scooter Doppler frequency estimation and detected frequency for bit 1.

**Figure 19 sensors-23-02730-f019:**
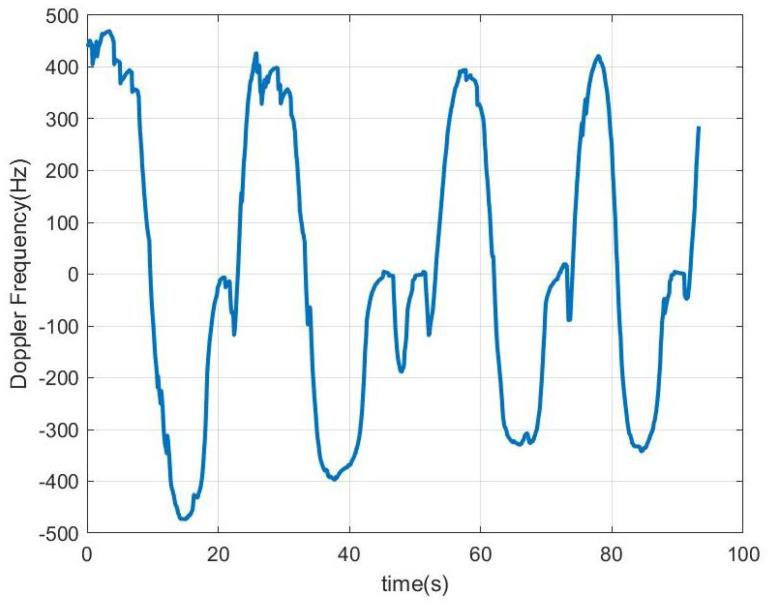
Measurement for a moving scooter sending Barker codes with length 7: scooter Doppler frequency estimation.

**Figure 20 sensors-23-02730-f020:**
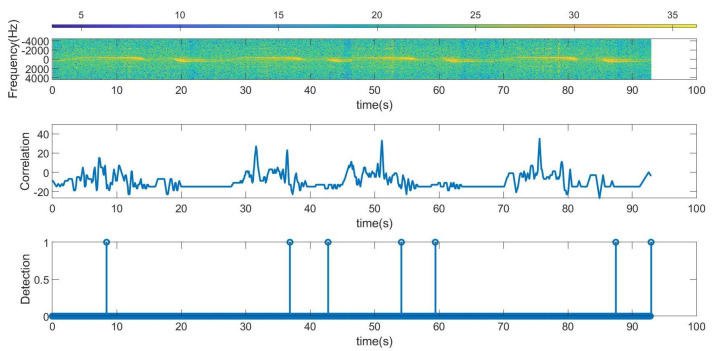
Measurement for a moving scooter sending Barker codes with length 13. From top to bottom: raw spectrogram before the CFAR processor is applied, the output of the correlator and tag detection.

**Figure 21 sensors-23-02730-f021:**
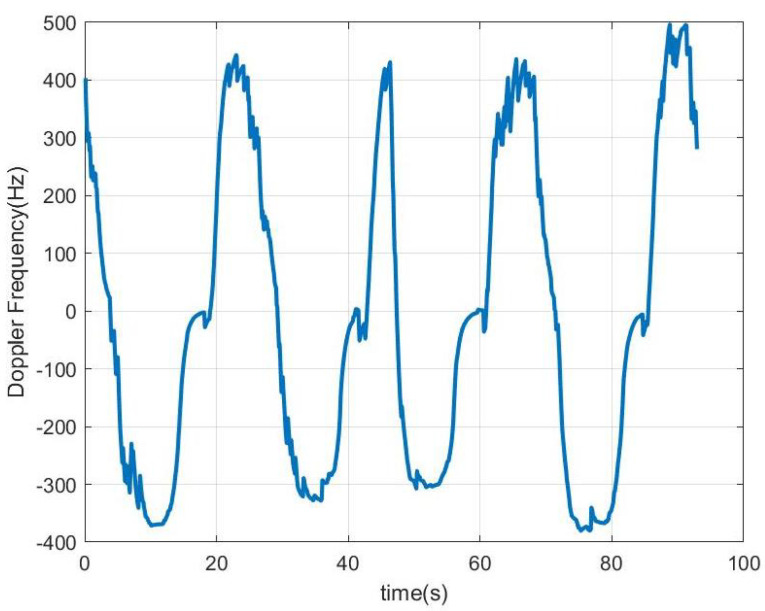
Measurement for a moving scooter sending Barker codes with length 13: Doppler frequency of the scooter estimation.

**Figure 22 sensors-23-02730-f022:**
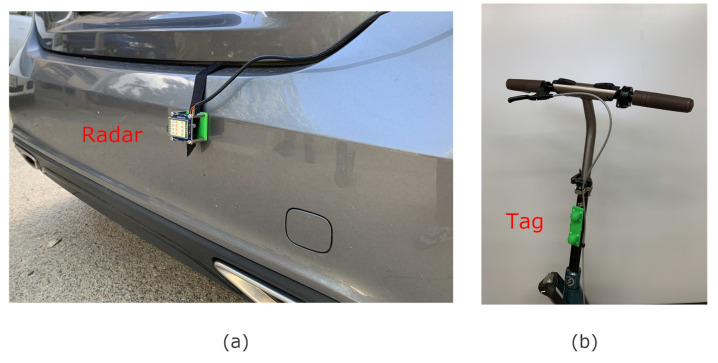
(**a**) Photograph of the radar installed at the rear of the car; (**b**) tag installed on the scooter.

**Figure 23 sensors-23-02730-f023:**
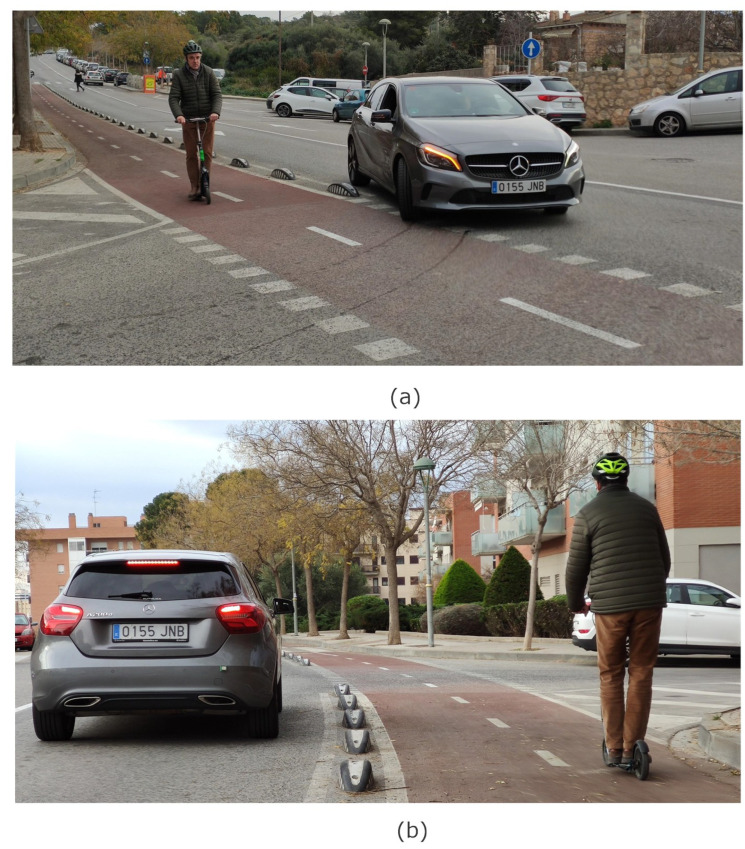
Photograph of the experimental setup (**a**) car changing direction; (**b**) turning car encroaching on the bike lane.

**Figure 24 sensors-23-02730-f024:**
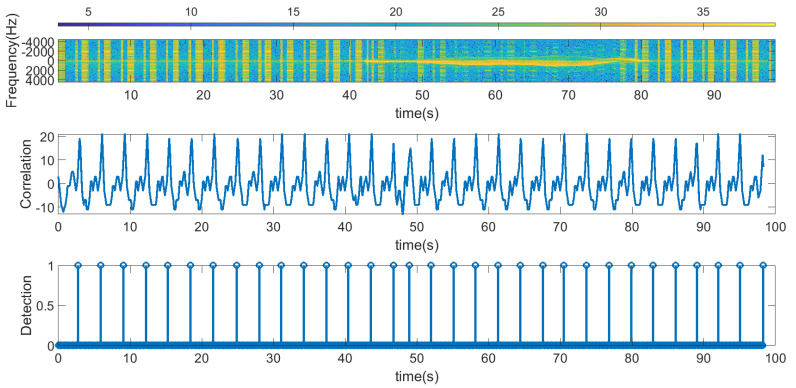
Data measurement using length 7 Barker codes when the scooter is in motion. From top to bottom: raw spectrogram before CFAR processor is applied, the output of the correlator signal and tag detection. The scooter is within the radar reading range at any time.

**Figure 25 sensors-23-02730-f025:**
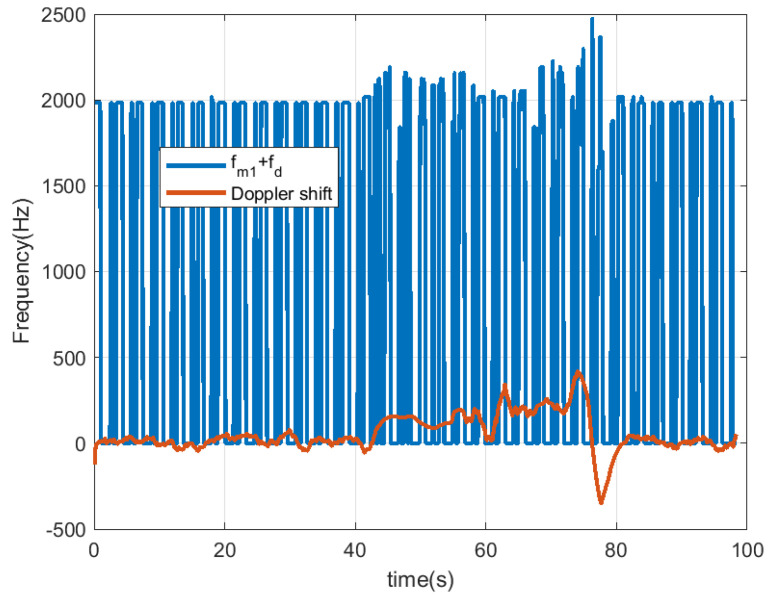
Data measurement using length 7 Barker codes when the scooter is in motion: scooter Doppler frequency estimate and detected frequency for bit 1. The scooter is within the radar reading range at any time.

**Figure 26 sensors-23-02730-f026:**
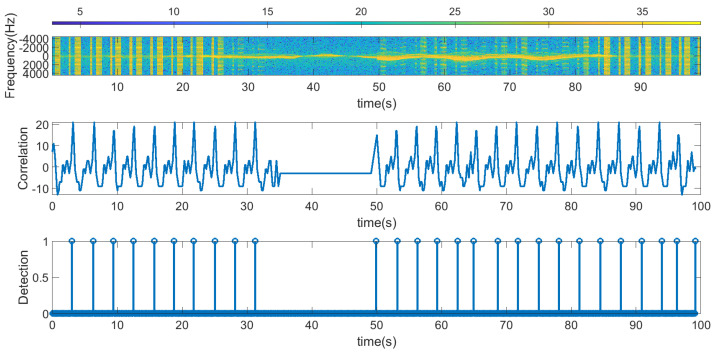
Data measurement using length 7 Barker codes when the scooter is in motion. From top to bottom: raw spectrogram before CFAR processor is applied, the output of the correlator signal and tag detection. In this case, the tag is partially out of range.

**Figure 27 sensors-23-02730-f027:**
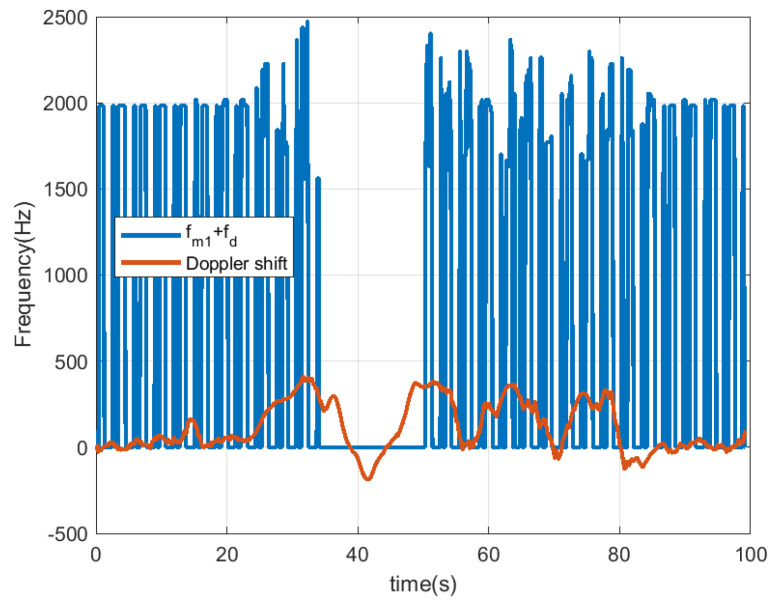
Data measurement using length 7 Barker codes when the scooter is in motion: scooter Doppler frequency estimate and detected frequency for bit 1. In this case, the tag is partially out of range.

**Figure 28 sensors-23-02730-f028:**
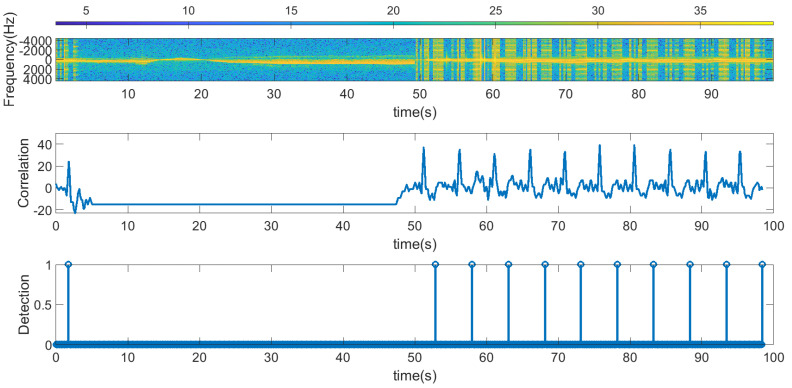
Data measurement using length 13 Barker codes when the scooter is in motion. From top to bottom: raw spectrogram before CFAR processor is applied, the output of the correlator signal and tag detection. In this case, the tag is partially out of range.

**Figure 29 sensors-23-02730-f029:**
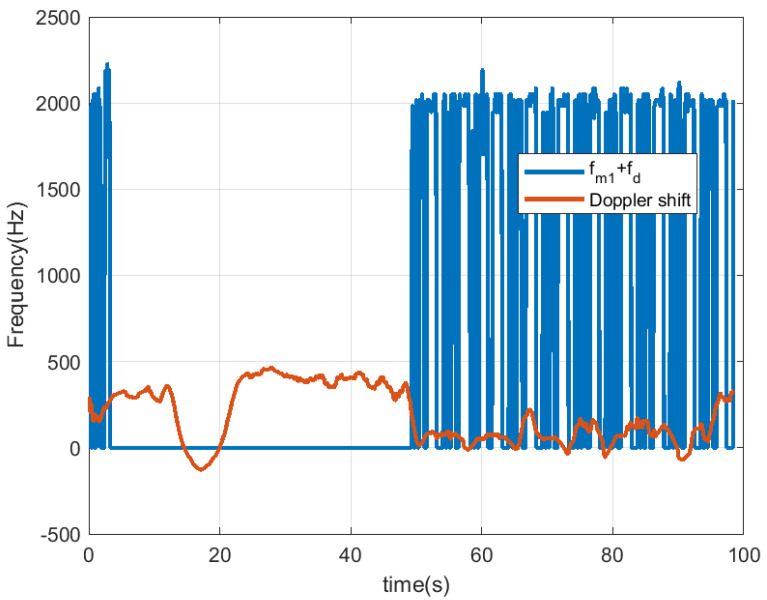
Data measurement using length 13 Barker codes when the scooter is in motion: scooter Doppler frequency estimate and detected frequency for bit 1. In this case, the tag is partially out of range.

**Table 1 sensors-23-02730-t001:** Barker codes and their auto-correlation sidelobe level ratio.

Length	Code	Sidelobe Ratio (dB)
2	+1 −1,+1 +1	−6
3	+1 +1 −1	−9.5
4	+1 +1 −1 +1,+1 +1 +1 −1	−12
5	+1 +1 +1 −1 +1	−14
7	+1 +1 +1 −1 −1 +1 −1	−16.9
11	+1 +1 +1 −1 −1 −1 +1 −1 −1 +1 −1	−20.8
13	+1 +1 +1 +1 +1 −1 −1 +1 +1 −1 +1 −1 +1	−22.3

**Table 2 sensors-23-02730-t002:** Summary of measured cases.

Figure	Tag	Radar	Code Used	Environment
Figure 16	Scooter, stationary	Stationary	Barker 7	Outdoor at 8 m
Figure 17	Scooter, stationary	Stationary	Barker 13	Outdoor at 8 m
Figure 18	Scooter in motion	Stationary	Barker 7	Outdoor
Figure 20	Scooter in motion	Stationary	Barker 13	Outdoor
Figure 24	Scooter in motion	Car in motion	Barker 7	Street within read range
Figure 26	Scooter in motion	Car in motion	Barker 7	Street partially out of read range
Figure 28	Scooter in motion	Car in motion	Barker 13	Street partially out of read range

## Data Availability

The data presented in this study are available from the corresponding author upon request.

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
