# Peer review of "Smart Spread Spectrum Modulated Tags for Detection of Vulnerable Road Users with Automotive Radar"

_sensors, 2023, doi:10.3390/s23052730_

Round 1

Reviewer 1 Report

The paper is interesting and clear. I have just one observation that, if I am not wrong, I would like the authors to point out in the paper. It is the fact that you need, at least, two tags installed in order to be 'visible' in an omnidirectional sense.

Reviewer 2 Report

1)      The abstract is quite short. The authors need to rewrite the abstract with stressing on motivation, comparison to recent works, and provide a summary of the main findings

2)      Introduction: this needs to be written in an organized way. The authors switch between “in this paper” and literature review. I recommend to split this section to two subsections: related works and contributions. Write the main contributions in bullets.

3)      eq. (1), what is pwm(t), is this the same as s(t). A clarification is needed.   

4)      Include the effect of wireless channel in the mathematical model,

5)      Provide more inside about figures 14 and 15

6)     Enlarge figures 16 and 17

7)      In line 444, explain why the use of longer codes requires a longer tag visibility time and therefore higher latency.

Round 2

Reviewer 2 Report

thank you